synthetic biology/computational biology/ applied mathematics

genetic code, codon, amino acid, mutation

**Author for correspondence:**
Paweł Błażej
e-mail: pawel.blazej@uwr.edu.pl

# Basic principles of the genetic code extension

Paweł Błażej, Małgorzata Wnetrzak, Dorota Mackiewicz and Paweł Mackiewicz

Department of Bioinformatics and Genomics, Faculty of Biotechnology, University of Wrocław, ul. Joliot-Curie 14a, Wrocław, Poland

PB, 0000-0003-3610-2765; PM, 0000-0003-4855-497X

Compounds including non-canonical amino acids (ncAAs) or other artificially designed molecules can find a lot of applications in medicine, industry and biotechnology. They can be produced thanks to the modification or extension of the standard genetic code (SGC). Such peptides or proteins including the ncAAs can be constantly delivered in a stable way by organisms with the customized genetic code. Among several methods of engineering the code, using non-canonical base pairs is especially promising, because it enables generating many new codons, which can be used to encode any new amino acid. Since even one pair of new bases can extend the SGC up to 216 codons generated by a six-letter nucleotide alphabet, the extension of the SGC can be achieved in many ways. Here, we proposed a stepwise procedure of the SGC extension with one pair of non-canonical bases to minimize the consequences of point mutations. We reported relationships between codons in the framework of graph theory. All 216 codons were represented as nodes of the graph, whereas its edges were induced by all possible single nucleotide mutations occurring between codons. Therefore, every set of canonical and newly added codons induces a specific subgraph. We characterized the properties of the induced subgraphs generated by selected sets of codons. Thanks to that, we were able to describe a procedure for incremental addition of the set of meaningful codons up to the full coding system consisting of three pairs of bases. The procedure of gradual extension of the SGC makes the whole system robust to changing genetic information due to mutations and is compatible with the views assuming that codons and amino acids were added successively to the primordial SGC, which evolved minimizing harmful consequences of mutations or mistranslations of encoded proteins.

# 1. Introduction

The basic diversity of proteins fulfilling a wide range of functions within organisms is based on 20 naturally occurring amino acids.

The proteins are also modified post-translationally, which extends their properties. However, it is tempting to increase this variety with artificially designed amino acids or other molecules. They can be introduced directly into proteins or modified in a given proteinaceous molecule, but a more universal and stable solution is such modification of the standard genetic code (SGC) that the newly created proteins including non-canonical amino acids (ncAAs) are constantly produced by a given organism. Several approaches were invented to achieve this goal [1].

The first approach uses stop translation codons (e.g. rarely used UAG) to encode ncAAs [2–5]. This method requires a modified aminoacyl-tRNA synthetase which charges a tRNA with an ncAA. This suppressor tRNA must recognize the stop codon and then ncAA is incorporated into a protein during its synthesis. However, this method enables utilization of up to two stop codons because one of the three codons must be left as a termination signal of translation [6].

Another method applies quadruplet codons, which consist of an infrequently used triplet codon with an additional base [7–9]. Such a quadruplet is decoded by a modified tRNA containing a complementary quadruplet anticodon. Then, ncAA associated with this tRNA is added into a newly synthesized protein due to frameshifted open reading frame. However, the typical triplet can be decoded by a typical tRNA competitively, which decreases the efficiency of this procedure.

It is also possible to assign various sense codons to different ncAAs by withdrawing the cognate amino acid and aminoacyl-tRNA synthetase, and adding pre-charged ncAA-tRNAs bearing the corresponding anticodons [10–12]. This method, however, sacrifices a natural amino acid. A new method overcomes this problem and frees sense codons for ncAAs without elimination of natural ones [13]. This is achieved by utilization of appropriate synonymous codons, depletion of their corresponding tRNAs and addition of tRNAs pre-charged with ncAAs. This method enables expanding the repertoire to 23 potential ncAAs via division of multiple codon boxes [14] but can influence the efficiency and speed of translation as well as protein folding due to altered codon usage [15].

A weakness of these methods is that they rely on the set of four canonical bases, which can generate a limited set of codons, up to 64. Therefore, a promising approach is using unnatural base pairs, which can generate a much larger number of genuinely new codons. This approach does not interfere with the natural system because it does not involve the canonical codons, while the new ones are free of any natural role. Such experiments with at least three pairs of the fifth and the sixth nucleotide were already carried out and appeared promising [16–22]. Protein synthesis using this approach occurred successfully in semi-synthetic bacteria [23].

The inclusion of one pair of unnatural nucleotides can extend the SGC even up to 216 codons, which is nearly three times larger than the set of 64 canonical codons. The 152 new unassigned codons raise an exciting possibility of adding many unnatural amino acids or similar compounds and creating a new extended genetic code (EGC). Therefore, it is reasonable to pose a question about the rules according to which we can extend the code. There are many possibilities to do this. Here, we propose a way assuming that the genetic code should be a system resistant to point mutations, which can change the encoded information. In other words, we present a formal description of the genetic code expansion to minimize the cost of changing codons due to the mutations. The presented procedure of incremental expansion of the genetic code ensures robustness of the extended code against losing genetic information. This assumption seems attractive in the context of the hypothesis postulating that the genetic code evolved to minimize harmful consequences of mutations or mistranslations of coded proteins [24–33].

# 2. Methods

## 2.1. The extension of the standard genetic code

We start our investigation by applying a similar approach to that presented by [34], in which the SGC is described as a graph $G(V_0, E_0)$, where $V_0$ is the set of vertices (nodes), whereas $E_0$ is the set of edges. $V_0$ corresponds to the set of 64 canonical codons using four natural nucleotides $\{A, T, G, C\}$, while the edges are induced by all possible single nucleotide substitutions between the codons. Therefore, the graph $G(V_0, E_0)$ is a representation of all possible single-point mutations occurring between canonical codons.

In this work, we introduce a more general graph $G(V, E)$, in which the set of vertices corresponds to 216 codons, using a six-letter alphabet, while the set of edges is defined in a similar way as $E_0$.

**Definition 2.1.** Let $G(V, E)$ be a graph in which $V$ is the set of vertices representing all possible 216 codons, whereas $E$ is the set of edges connecting these vertices. All connections between the nodes fulfil the property that two nodes, i.e. codons $u, v \in V$, are connected by the edge $e(u, v) \in E$ ($u \sim v$), if and only if the codon $u$ differs from the codon $v$ in exactly one position.

In order to simplify our notation, we use further $G$ instead of $G(V, E)$. It is clear that the set of edges $E$ of the graph $G$ represents all possible single nucleotide substitutions, which occur between codons created by the set of natural nucleotides $\{A, T, G, C\}$ as well as one pair of unnatural nucleotides $\{X, Y\}$. Assuming that all changes are equally probable, we obtain that $G$ is an undirected, unweighted and regular graph with the vertices degree equal to 15. Moreover, the set of 64 canonical codons $V_0$ used in the SGC is a subset of $V$. Therefore, $V_0$ induces a subgraph $G[V_0]$ of the graph $G(V, E)$ according to the following definition.

**Definition 2.2.** If $G(V, E)$ is a graph, and $S \subset V$ is a subset of vertices of $G$, then the induced subgraph $G[S]$ is the graph whose set of vertices is $S$ and whose set of edges consists of the edges in $E$, which have both endpoints in $S$.

Following this definition, let us denote by $V_n$ a subset of vertices (codons) involved in a given EGC with exactly $n \geq 1$ non-canonical codons. This subset must fulfil the following property:

$$V_0 \subset V_n \subseteq V,$$

i.e. $V_n$ must be an extension of the set of canonical codons. As a result, we can define a graph $G[V_n]$, which is a subgraph of the graph $G$ generated by $V_n$. Therefore, the main goal of this work is to test the property of the graph $G[V_n]$, which can be interpreted as a structural representation of the EGC. Thus, we develop methodology to describe features of the graph $G$.

## 2.2. The properties of the graph $G$

Interesting features of $G$ can appear, when the set of vertices $V$ is divided into the partition of eight disjoint and non-empty sets. It induces a specific connection between these vertices by edges. This partition includes also $V_0$, i.e. the set of natural codons.

**Proposition 2.3.** *Let $G(V, E)$ be a graph, where $V$ represents the set of all possible $216$ codons and $E$ is the set of edges generated by single nucleotide substitutions. Then, the set of vertices $V$ can be split unambiguously into eight disjoint subsets. These are $V_0$, $B_1$, $B_2$, $B_3$, $B_{12}$, $B_{13}$, $B_{23}$ and $B_{123}$, where*

(a) *$V_0$ is the set of 64 canonical codons;*
(b) *$B_1$ is the set of codons in which new nucleotides $X$ or $Y$ occur only in the first codon position;*
(c) *$B_2$ is the set of codons in which new nucleotides $X$ or $Y$ occur only in the second codon position;*
(d) *$B_3$ is the set of codons in which new nucleotides $X$ or $Y$ occur only in the third codon position;*
(e) *$B_{12}$ is the set of codons in which new nucleotides $X$ or $Y$ occur in the first and the second codon position;*
(f) *$B_{13}$ is the set of codons in which new nucleotides $X$ or $Y$ occur in the first and the third codon position;*
(g) *$B_{23}$ is the set of codons in which new nucleotides $X$ or $Y$ occur in the second and the third codon position;*
(h) *$B_{123}$ is the set of codons in which new nucleotides $X$ or $Y$ occur in all codon positions.*

The number of elements, i.e. codons in theses sets are: $|B_1| = |B_2| = |B_3| = 32$, $|B_{12}| = |B_{23}| = |B_{13}| = 16$ and $|B_{123}| = 8$.

The graphical of relationships between these sets is presented in figure 1.

Based on such partition, we can investigate properties of the EGC. In order to do this, let us introduce the following notation. We denote another three subsets of $V$

$$C^1 = V_0 \cup B_1 \cup B_2 \cup B_3, \tag{2.1}$$

The sets $C^1$ and $B_{12} \cup B_{13} \cup B_{23} \cup B_{123}$ are disjoint and constitute also a partition of $V$. We call the set $B_1 \cup B_2 \cup B_3$ 'close neighbourhood' of $V_0$ because it contains all codons that differ from the set $V_0$ in at most one position in a codon. In contrast to that, $B_{12} \cup B_{13} \cup B_{23} \cup B_{123}$ is not directly connected with $V_0$. Moreover, we introduce also the set $C^2$, defined as follows:

$$C^2 = C^1 \cup B_{12} \cup B_{23} \cup B_{13}. \tag{2.2}$$

It is clear that $C^2$ and $B_{123}$ are disjoint and also constitute a partition of $V$.

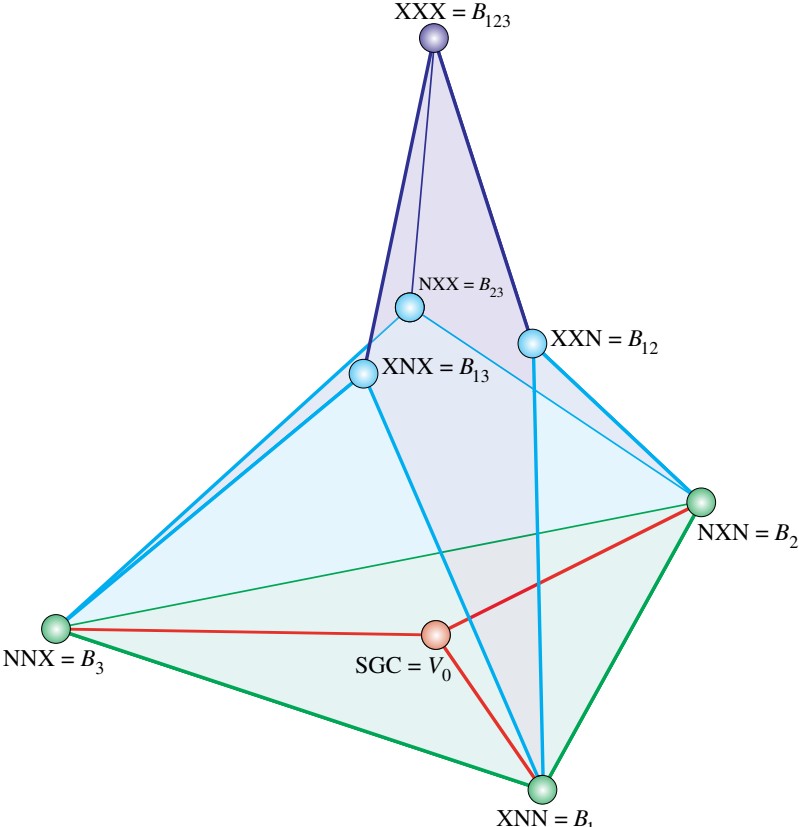

**Figure 1.** The graphical of extended genetic code (EGC) based on the standard genetic code (SGC). The graph is induced by the partition of the set of vertices $V$ from the graph $G$ into eight subsets $V_0$, $B_1$, $B_2$, $B_3$, $B_{12}$, $B_{13}$, $B_{23}$ and $B_{123}$, where the edges between the groups are induced by edges connecting codons, which belong to different sets. The edges correspond to single-point mutations between the codons. $V_0$ corresponds to the SGC, N refers to a standard base and X a non-standard base.

In the next proposition, we give several properties of edge connections between the selected sets of nodes.

**Proposition 2.4.** *Let us consider the codon sets introduced in proposition 2.3 and two subsets of nodes $C^1$, $C^2$. Then we have the following properties*:

(a) *Each codon $c \in B_i$, $i = 1, 2, 3$ has exactly four edges crossing from $B_i$ to $V_0$;*
(b) *Each codon $c \in B_i$, $i = 1, 2, 3$ has exactly four edges crossing from $X_i$ to $B_{12} \cup B_{13} \cup B_{23} \cup B_{123}$;*
(c) *There does not exist any connection between $B_{12} \cup B_{13} \cup B_{23} \cup B_{123}$ and $V_0$;*
(d) *Each codon $c \in B_{ij}$, $i \neq j$, $i, j = 1, 2, 3$ has exactly eight edges crossing to $C^1$;*
(e) *Each codon $c \in B_{ij}$, $i = 1, 2, 3$ has exactly two edges crossing from $B_{ij}$ to $B_{123}$;*
(f) *There does not exist any connection between $B_{ij}$ and $V_0$;*
(g) *There does not exist any connection between $B_{123}$ and $C^1$.*

It is also interesting to describe some properties of subgraphs generated by codon sets $B_1$, $B_2$, $B_3$ and $B_{12}$, $B_{13}$, $B_{23}$, respectively. They are formulated in the following two lemmas:

**Lemma 2.5.** *Graphs $G[B_1]$, $G[B_2]$ and $G[B_3]$ are isomorphic to each other.*

*Proof.* According to the definition of the graph isomorphism, there must exist a bijection $f$ between $G[B_i]$ and $G[B_j]$, $i \neq j$, i.e. $f : G[B_i] \to G[B_j]$ such that two vertices $u$, $v$ are adjacent in $G[B_i]$, if and only if $f(u)$ and $f(v)$ are adjacent in $G[B_j]$. In this case, such a bijection can be easily defined as a swap between respective codon positions, where nucleotides $X$ and $Y$ occur. ∎

We observe the same property in the case of codon sets $B_{12}$, $B_{13}$ and $B_{23}$. Thus, we can formulate a similar lemma:

**Lemma 2.6.** *Graphs $G[B_{12}]$, $G[B_{13}]$ and $G[B_{23}]$ are isomorphic to each other.*

*Proof.* The proof is analogous to the proof of lemma 2.5.

What is more, in the construction of the optimal EGC, we also use the fact that the graphs $G[B_n]$, $n \in \{1, 2, 3, 12, 13, 23, 123\}$, can be represented as Cartesian products of other graphs. This important feature is presented in the following three propositions.

**Proposition 2.7.** *The graph $G[B_1]$, can be represented as a Cartesian product of graphs:*

$$G[B_1] = K_2 K_4 K_4,$$

*where $K_2$ and $K_4$ are complete graphs of sizes two and four with the set of vertices $\{X, Y\}$ and $\{A, T, G, C\}$, respectively. In this case, two vertices $(x, y, z)$, $(x', y', z')$ are connected by the edge $e((x, y, z), (x', y', z'))$, if $(x = x'$ and $y = y'$ and $z \sim z')$ or $(x = x'$ and $y \sim y'$ and $z = z')$ or $(x \sim x'$ and $y = y'$ and $z = z')$.*

**Proposition 2.8.** *The graph $G[B_{12}]$, can be represented as a Cartesian product of graphs*

$$G[B_{12}] = K_2 K_2 K_4,$$

*where $K_2$ and $K_4$ are complete graphs of sizes two and four with the set of vertices $\{X, Y\}$ and $\{A, T, G, C\}$, respectively. In this case, two vertices $(x, y, z)$, $(x', y', z')$ are connected by the edge $e((x, y, z), (x', y', z'))$, if $(x = x'$ and $y = y'$ and $z \sim z')$ or $(x = x'$ and $y \sim y'$ and $z = z')$ or $(x \sim x'$ and $y = y'$ and $z = z')$.*

**Proposition 2.9.** *The graph $G[B_{123}]$ can be represented as a Cartesian product of graphs*

$$G[B_{123}] = K_2 K_2 K_2,$$

*where $K_2$ is a complete graph of size two with the set of vertices $\{X, Y\}$. In this case, two vertices $(x, y, z)$, $(x', y', z')$ are connected by the edge $e((x, y, z), (x', y', z'))$, if $(x = x'$ and $y = y'$ and $z \sim z')$ or $(x = x'$ and $y \sim y'$ and $z = z')$ or $(x \sim x'$ and $y = y'$ and $z = z')$.*

## 2.3. The optimality of codon group

Similarly to [34], we introduce two measures describing properties of codon groups. They are the set conductance and the $k$-size conductance, which characterize the quality of a given codon sets in terms of non-synonymous mutations which lead to a replacement of one amino acid by another.

**Definition 2.10.** For a given graph $G$, let $S$ be a subset of $V$. The conductance of $S$ is defined as

$$\phi(S) = \frac{E(S, \bar{S})}{vol(S)},$$

where $E(S, \bar{S})$ is the number of edges of $G$ crossing from $S$ to its complement $\bar{S}$ and $vol(S)$ is the sum of all degrees of the vertices belonging to $S$.

The measure $\phi(S)$ can be interpreted as a fraction of non-synonymous substitutions between $S$ and $\bar{S}$, if $S$ is a group of codons encoding the same amino acid and $\bar{S}$ includes codons bearing other genetic information. It is interesting that the optimal codon group, in terms of its robustness to point mutations, should be characterized by low values of the set conductance. Therefore, the number of nucleotide substitutions that change a coded amino acid should be relatively small in comparison to the total number of all possible nucleotide mutations involving all codons belonging to the given set. In this context, it is also interesting to calculate the $k$-size-conductance $\phi_k(G)$, which is described as the minimal set conductance over all subsets of $V$ with the fixed size $k$.

**Definition 2.11.** The $k$-size-conductance of the graph $G$, for $k \geq 1$, is defined as:

$$\phi_k(G) = \min_{S \subseteq V, |S|=k} \phi(S).$$

In consequence, $k \cdot \phi_k(G)$ gives us a lower bound on the number of edges going outside the set nodes of the size $k$ and this characteristic is useful in describing the optimal codon structures.

## 3. Results

In this section, we present a step by step procedure which allows us to extend the SGC from 64 up to 216 meaningful codons. Codons are added to the code gradually in three stages. The first step extends the

SGC to 160 codons, the second step to 208 codons and the third to all possible 216 codons. The EGC created at each stage must be optimal in terms of minimization of point mutations.

## 3.1. The optimal extension of the standard genetic code to 160 meaningful codons

Following the properties of the graph $G$, we formulate some characteristics, which are useful in describing the properties of the subgraph $G[V_n]$, $1 \leq n \leq 96$ induced by the set of codons $V_n$ and at the same time in developing the optimal EGC. At the beginning, we propose some optimization criteria in order to find the best possible solution.

Using the notation from the previous sections, let us define

$$\overline{V_n} = V \backslash V_n,$$

which is a set of unassigned codons. Moreover, let us denote by $A_n$ a set of $n$ new codons involved in a given genetic code extension

$$A_n = V_n \backslash V_0,$$

where $1 \leq |A_n| \leq 96$. Thanks to that, we can define two measures describing the properties of $G[V_n]$. They are

$$E(V_0, A_n) \tag{3.1}$$

and

$$E(V_n, \overline{V_n}), \tag{3.2}$$

where $E(V_0, A_n)$ is the total number of edges, extracted from the graph $G$, crossing from the set of canonical codons $V_0$ to $A_n$, whereas $E(V_n, \overline{V_n})$ is the total number of edges crossing from the set of codons which constitute the EGC $V_n$ to unassigned codons.

Interestingly, by applying (3.1) and (3.2), it is possible to characterize the properties of a given subgraph $G[V_n]$ and at the same time the EGC induced by the codons belonging to $V_n$. In the definition below, we give some conditions which constitute the EGC optimality. Thanks to that, we can find the best genetic code extended by $1 \leq n \leq 96$ new codons.

**Definition 3.1.** The set $V_n^*$, $V_0 \subset V_n^*$ with exactly $1 \leq n \leq 96$ non-canonical codons is an optimal extension of SGC, if

$$V_n^* = \underset{\{V_n:\ V_n = V_0 \cup A_n\}}{\arg\min}\ E(V_n, \overline{V_n}), \tag{3.3}$$

where $A_n$ possesses the feature

$$A_n = \underset{\{S:\ S \subseteq \overline{V_0},\ |S| = n\}}{\arg\max}\ E(V_0, S). \tag{3.4}$$

These two restrictions have a sensible interpretation. By minimizing the condition (3.3), we reduce the possibility that a point mutation can generate a codon belonging to the 'non-coding zone' $\overline{V_n}$, i.e. the set of unassigned codons. On the other hand, maximizing the value of $A_n$ according to (3.4), we claim that the number of connections between two sets, namely, the canonical and newly assigned codons $E(V_0, A_n)$ is as large as possible (figure 2).

These two assumptions maximize the number of connections between standard and newly incorporated codons and simultaneously decrease the probability of losing genetic information from the whole system due to point mutations. Therefore, we focus on the $V_n$ sets, when $A_n = V_n \backslash V_0$ fulfils the property (3.4). Then, let us denote by

$$\mathcal{V}_n = \{V_n:\ V_n = V_0 \cup A_n\}$$

a class of all sets $V_n$ with exactly $n$ non-canonical codons and let us assume that $A_n = V_n \backslash V_0$ fulfils the property (3.4). It is clear that all optimal EGCs, in terms of (3.4) and (3.3), belong to $\mathcal{V}_n$.

These features appear to be very useful for characterizing possible extensions of the SGC. In the next theorem, we describe the optimal extension of the SGC up to 160 meaningful codons. Interestingly, this extension can be described in terms of $k$-size conductance $\phi_k(G[B_i])$, $i = 1, 2, 3$ calculated for induced subgraphs $G[B_i]$, $i = 1, 2, 3$. We begin our investigation with a lemma, which gives us some characterizations of the optimal sets $V_n$.

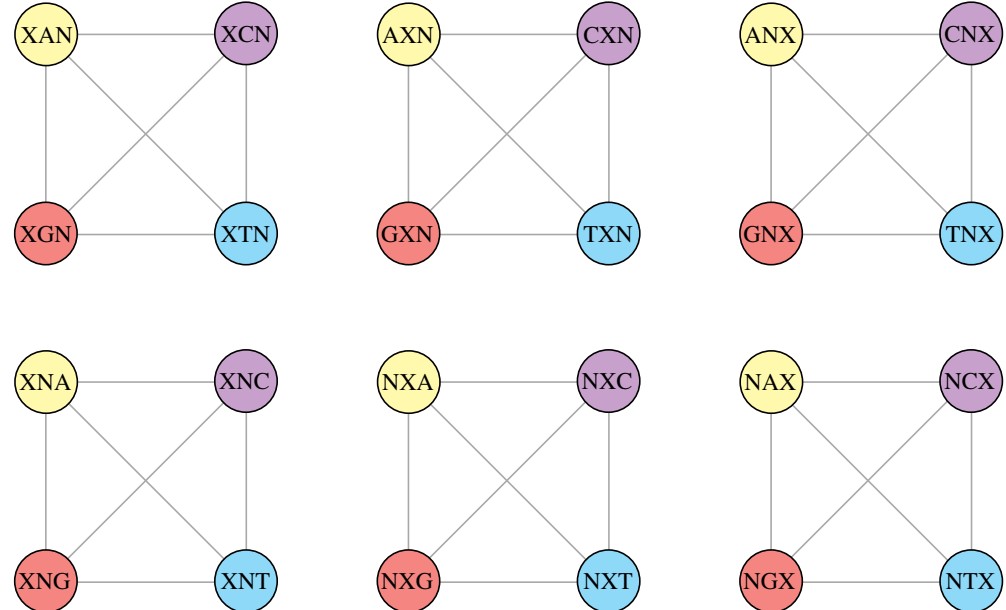

**Figure 2.** Examples of the optimal set of four new codons $A_4$ involved in a genetic code extension up to 160 codons.

**Lemma 3.2.** *Let $V_n \in \mathcal{V}_n$ be a set of codons, where $1 \leq n \leq 96$ and $A_n = V_n \backslash V_0$, then*

$$A_n \subseteq B_1 \cup B_2 \cup B_3.$$

*Proof.* The proof of this lemma follows directly from proposition 2.4(a,c) and the definition of $\mathcal{V}_n$. ∎

Thanks to that, we can formulate a theorem, which gives us a lower bound on the number of edges crossing from $V_n$ to its complement.

**Theorem 3.3.** *Let $V_n \in \mathcal{V}_n$ be a set of codons, where $1 \leq n \leq 96$. Then the following inequality holds*:

$$E(V_n, \overline{V_n}) \geq E(V_0, \overline{V_0}) + \sum_{i=1}^{3} n_i \cdot \phi_{n_i}(G[B_i]) = 384 + \sum_{i=1}^{3} n_i \cdot \phi_{n_i}(G[B_i]),$$

*where $G[B_i]$, $i = 1, 2, 3$ is the induced subgraph of $G$, and $n_i = |A_n \cap B_i|$, $n_1 + n_2 + n_3 = n$.*

*Proof.* We begin the proof with an observation

$$E(V_n, \overline{V_n}) = E(V_0, \overline{V_0}) - E(V_0, A_n) + E(A_n, \overline{V_n}). \tag{3.5}$$

Interestingly, following the definition 2.10, we can calculate the set conductance of $V_0$. In this case, we have $\phi(V_0) = 0.4$. Hereby, we get immediately

$$E(V_0, \overline{V_0}) = 64 \cdot 15 \cdot \phi(V_0) = 384. \tag{3.6}$$

In addition, using proposition 2.4(b) we get the following equality:

$$E(V_0, A_n) = 4n.$$

Therefore, we can rewrite the equality (3.5) in the following way:

$$E(V_n, \overline{V_n}) = 384 - 4n + E(A_n, \overline{V_n}). \tag{3.7}$$

In our next step, we observe

$$E(A_n, \overline{V_n}) = \sum_{i=1}^{3} E(A_n \cap B_i, \overline{V_n} \cap B_i) + \sum_{i=1}^{3} E(A_n \cap B_i, B_{12} \cup B_{13} \cup B_{23} \cup B_{123}),$$

where $\sum_{i=1}^{3} E(A_n \cap B_i, B_{12} \cup B_{13} \cup B_{23} \cup B_{123}) = 4n$ according to proposition 2.4(b). As a consequence,

we can reformulate equation (3.7) as follows:

$$E(V_n, \overline{V_n}) = 384 - 4n + \sum_{i=1}^{3} E(A_n \cap X_i, \overline{V_n} \cap B_i) + 4n.$$

Furthermore, taking into account that the set $\overline{V_n} \cap B_i = B_i \backslash (A_n \cap B_i)$ and using definitions 2.10 and 2.11, we have

$$\sum_{i=1}^{3} E(A_n \cap B_i, \overline{V_n} \cap B_i)$$

$$\geq \sum_{i=1}^{3} \min_{S \subseteq B_i, |S|=n_i} \frac{E(S, \overline{S})}{\mathrm{vol}(S)} \cdot \mathit{vol}(S)$$

$$\geq \sum_{i=1}^{3} \phi_{n_i}(G[B_i]) \cdot n_i.$$

Finally, we obtain

$$E(V_n, \overline{V_n}) \geq 384 + \sum_{i=1}^{3} n_i \cdot \phi_{n_i}(G[B_i]). \tag{3.8}$$

∎

Therefore, to extend the SGC using $1 \leq n \leq 96$ codons in the optimal way according to the definition 3.1, we have to choose codons only from the sets $B_1$, $B_2$ and $B_3$. Interestingly, the lower bound on the value of $E(V_n, \overline{V_n})$ presented in this theorem depends on the $n$-size conductance of new codon groups. What is more, the EGC being optimal in terms of the definition 3.1 and including 160 codons is described by the set $C^1$ because in this case we get

$$E(C^1, \overline{C^1}) = E(V_0, \overline{V_0}) = 384. \tag{3.9}$$

## 3.2. The properties of the optimal genetic code including up to 160 meaningful codons

We pose a question about the properties of the optimal codon set for which the lower bound

$$E(V_n, \overline{V_n}) = 384 + \sum_{i=1}^{3} n_i \cdot \phi_{n_i}(G[B_i])$$

is attained under the additional restriction $1 \leq n_1 + n_2 + n_3 \leq 96$, where $n_i$, $i = 1, 2, 3$ is the number of new codons introduced into EGC and belonging to $B_i$, $i = 1, 2, 3$, respectively. Moreover, it is also interesting to find the best possible genetic code extension for every $1 \leq n \leq 96$.

We begin our consideration with presenting some features of induced graphs $G[B_i]$, $i = 1, 2, 3$. These properties allow us to describe the optimal codon group in terms of $\phi_k(G[B_i])$. Following lemma 2.5, we get that $G[B_i]$, $i = 1, 2, 3$ are isomorphic to each other, hereby it is enough to consider the properties of the graph $G[B_1]$ (figure 3) because all potential code structures and also their properties can be transmitted unambiguously from $B_1$ to $B_2$ and $B_3$. Since the graph $G[B_1]$ has a representation as a Cartesian product of graphs (see proposition 2.7), in the light of theorem 2.3 from [35], we get that the collection of the first $n$ vertices of $G[B_1]$ taken in the lexicographic order is characterized by the set conductance values, which are optimal in terms of $k$-size conductance. Therefore, for every $V_n$, we can find a lower bound, i.e. an EGC which is composed of the subsets of lexicographically ordered codons belonging to $B_1 \cup B_2 \cup B_3$.

In table 1, we present the list of all $G[B_1]$ nodes taken in the selected lexicographic order. What is more, we evaluate also all possible $k$-size conductance values for the respective sets. Using these results, we can propose a method for finding the best possible genetic code extension in the class $V_n$, $1 \leq n \leq 96$. Let us start with the following observation: if $n_1$, $n_2$ and $n_3$ defined in theorem 3.3 fulfil the condition $n_1 + n_2 + n_3 \leq 32$, then we get the following inequality:

$$3 \cdot \min(\phi_{n_1}(G[B_i]), \phi_{n_2}(G[B_i]), \phi_{n_3}(G[B_i])) \geq \phi_{n_1+n_2+n_3}(G[B_i]).$$

This formula results from the fact that the calculated values of $\phi_n(G[B_1])$ decrease, in general, with the size of codon groups $n$ (table 1). Therefore, to create the optimal genetic code extension $V_n^*$, it is enough to

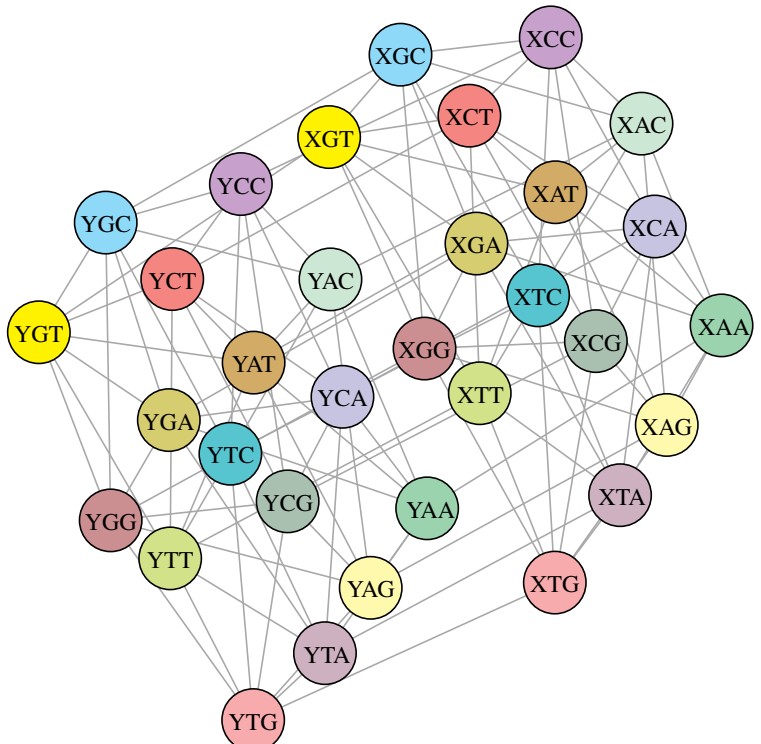

**Figure 3.** The graphical of the graph $G[B_1]$ which is an induced subgraph of the graph $G(V, E)$. Each node is a codon belonging to the set $B_1$, $B_1 \subset V$, whereas its edges are taken from the set $E$.

choose new codons from the set $B_i$ until the total number of codons $n$ exceeds 32. Then, this procedure should be continued and additional codons from the next $B_i$-type set should be selected until the total number of codons reaches 64.

## 3.3. The optimal extension of the standard genetic code with more than 160 meaningful codons

In order to extend the genetic code over 160 meaningful codons, we have to make some observations. From (3.9), we get immediately that $C^1$ is the best genetic code extension involving 96 additional codons. In addition, applying proposition 2.4(c), we get that $C^1$ includes all non-canonical codons that are directly connected with $V_0$. As a result, the condition (3.4) is non-restrictive in the case when we try to extend $V_0$ in consecutive steps using definition 3.1 for $n > 96$. Therefore, we propose to reformulate the problem of optimal $V_0$ extension into the question of optimal extension of the $C^1$ set.

Let us denote by $V'_n$ a set of codons such that $C^1 \subseteq V'_n$ with exactly $n$, $1 \leq n \leq 48$ new codons in comparison to $C^1$. Therefore, the optimal genetic code extension can be characterized in the following way.

**Definition 3.4.** The set $V'^{*}_n$, $C^1 \subset V'^{*}_n$ with exactly $n$ additional codons is optimal if

$$V'^{*}_n = \arg\min_{\{V'_n:\ V'_n = C^1 \cup A'_n\}} E(V'_n, \overline{V'_n}), \tag{3.10}$$

where $A'_n$ possess the feature

$$A'_n = \arg\max_{\{S:\ S \subseteq \overline{C^1},\ |S|=n\}} E(C^1, S). \tag{3.11}$$

Similarly to the method presented in the previous subsections, we introduce a definition which is useful in describing the optimality of the EGC.

**Definition 3.5.** Let us define by $\mathcal{V}'_n$ a class of sets $V'_n$, whose $1 \leq n \leq 48$ additional codons and $A'_n = V'_n \backslash C^1$ fulfil the property (3.11). Then,

$$\mathcal{V}'_n = \{V'_n:\ V'_n = C^1 \cup A'_n\},$$

is a class of all possible extensions of the $C^1$ set with exactly $n$ new codons.

**Table 1.** The sequence of codons composing the set $B_1$. They are ordered according to a selected lexicographic order. The values of the $k$-size conductance $\phi_k(G[B_1])$ calculated for the first $k$ codons in order are also presented.

| codon | $k$ | $\phi_k(G[B_1])$ |
|---|---|---|
| XAA | 1 | 1 |
| XAT | 2 | 0.857 |
| XAG | 3 | 0.714 |
| XAC | 4 | 0.571 |
| ATA | 5 | 0.600 |
| XTT | 6 | 0.571 |
| XTG | 7 | 0.510 |
| XTC | 8 | 0.428 |
| XGA | 9 | 0.428 |
| XGT | 10 | 0.400 |
| XGG | 11 | 0.350 |
| XGC | 12 | 0.285 |
| XCA | 13 | 0.274 |
| XCT | 14 | 0.244 |
| XCG | 15 | 0.200 |
| XCC | 16 | 0.143 |
| YAA | 17 | 0.176 |
| YAT | 18 | 0.190 |
| YAG | 19 | 0.188 |
| YAC | 20 | 0.171 |
| YTA | 21 | 0.184 |
| YTT | 22 | 0.182 |
| YTG | 23 | 0.168 |
| YTC | 24 | 0.143 |
| YGA | 25 | 0.143 |
| YGT | 26 | 0.132 |
| YGG | 27 | 0.111 |
| YGC | 28 | 0.082 |
| YCA | 29 | 0.074 |
| YCT | 30 | 0.057 |
| YCG | 31 | 0.032 |
| YCC | 32 | 0 |

Thanks to that, we are able to give the optimal $C^1$ extension with a given size $n$. In order to increase the SGC by over 160 codons in total, it is enough to extend the set $C^1$ by incorporating new codons from the set

$$B_{12} \cup B_{13} \cup B_{23} \cup B_{123},$$

in such a way that the number of connections between a new code and its complement is minimized according to the condition (3.10), whereas the number of possible connections between the 'basic' coding system $C^1$ and newly added codons is maximized at the same time according to the condition (3.11).

Interestingly, we can find the optimal $C^1$ extension for $1 \leq n \leq 48$ in a similar way to that presented in §3.1. We begin by introducing the following lemma.

**Lemma 3.6.** *Let $V'_n \in \mathcal{V}'_n$ be a set of codons where $1 \le n \le 48$ and $A'_n = V'_n \backslash C^1$. If $A'_n$ fulfils the condition (3.11), then*

$$A'_n \subseteq B_{12} \cup B_{13} \cup B_{23} \cup B_{123}.$$

*Proof.* The proof of this lemma follows directly from proposition 2.4(d,g). ∎

Then, we can formulate the following theorem.

**Theorem 3.7.** *Let $V'_n \in \mathcal{V}'_n$ be a set of codons, where $1 \le n \le 48$ and $A'_n = V'_n \backslash C^1$ fulfil the condition (3.11). Then the following inequality holds*:

$$E(V'_n, \overline{V'_n}) \ge 384 - 6n + \sum_{ij} n_{ij} \cdot \phi_{n_{ij}}(G[B_{ij}]),$$

*where $n_{ij} = |A'_n \cap X_{ij}|$, $\sum_{ij} n_{ij} = n$ and $G[B_{ij}]$ is the induced subgraph of $G$.*

*Proof.* Similarly to the proof of theorem 3.3, we start with the equation:

$$E(V'_n, \overline{V'_n}) = E(C^1, \overline{C^1}) - E(C^1, A'_n) + E(A'_n, \overline{V'_n}). \tag{3.12}$$

Using equation (3.9) and proposition 2.4(d,g), we get immediately two equalities

$$E(C^1, \overline{C^1}) = 384, \quad E(C^1, A'_n) = 8n.$$

Therefore, we can rewrite equation (3.12) in the following way:

$$E(V'_n, \overline{V'_n}) = 384 - 8n + E(A'_n, \overline{V'_n}).$$

In the next step, we make a simple observation

$$E(A'_n, \overline{V'_n}) = \sum_{ij} E(A'_n \cap B_{ij}, \overline{V'_n} \cap B_{ij}) + \sum_{ij} E(A'_n \cap B_{ij}, B_{123}),$$

where $\sum_{ij} E(A'_n \cap B_{ij}, B_{123}) = 2n$ according to proposition 2.4(e). Then following the definitions 2.10 and 2.11, we get:

$$\sum_{ij} E(A'_n \cap X_{ij}, \overline{V'_n} \cap B_{ij}) \ge \sum_{ij} n_{ij} \cdot \phi_{n_{ij}}(G[B_{ij}]).$$

In consequence, we can reformulate the inequality 3.12 as follows:

$$E(V'_n, \overline{V'_n}) \ge 384 - 6n + \sum_{ij} n_{ij} \cdot \phi_{n_{ij}}(G[B_{ij}]).$$

∎

As a result, we found the lower bound of the value of $E(V'_n, \overline{V'_n})$, where the size $n$ of the set $V'_n$ is a number between $1 \le n \le 48$. Similarly to theorem 3.3, the optimality of the EGC depends strongly on the properties of newly created codon groups. Clearly, the best codon groups attain the $n$-size conductance values $\phi_{n_{ij}}(G[B_{ij}])$ for their size $n_{ij}$. What is more, the optimal EGC, in terms of the definition 3.4 with 208 codons in total, is described by the set $C^2$ because in this case we get

$$E(C^2, \overline{C^2}) = 384 - 6 \cdot 48 = 96. \tag{3.13}$$

## 3.4. The optimal codon block structures including up to 208 meaningful codons

As was mentioned in the previous section, the properties of the newly incorporated codons have a decisive impact on the optimality of the EGC. Applying theorem 3.7, the lower limitation on the value of $E(V'_n, \overline{V'_n})$, under the condition (3.11), is determined by the codon blocks that are optimal in terms of the $k$-size conductance. Following the results presented in §3.2, we have to consider some properties of induced subgraphs $G[B_{ij}]$, $ij = 12, 13, 23$, because they allow us to describe the optimal codon groups. Using lemma 2.6, we obtain that graphs $G[B_{ij}]$ are isomorphic to each other. Thanks to that, it is sufficient to consider the properties of the graph $G[B_{12}]$ (figure 4). Similarly to the previous results, $G[B_{12}]$ can be represented as a Cartesian product of graphs(lemma 2.8). Therefore, using again theorem 2.3 from [35], we obtain that the set of the

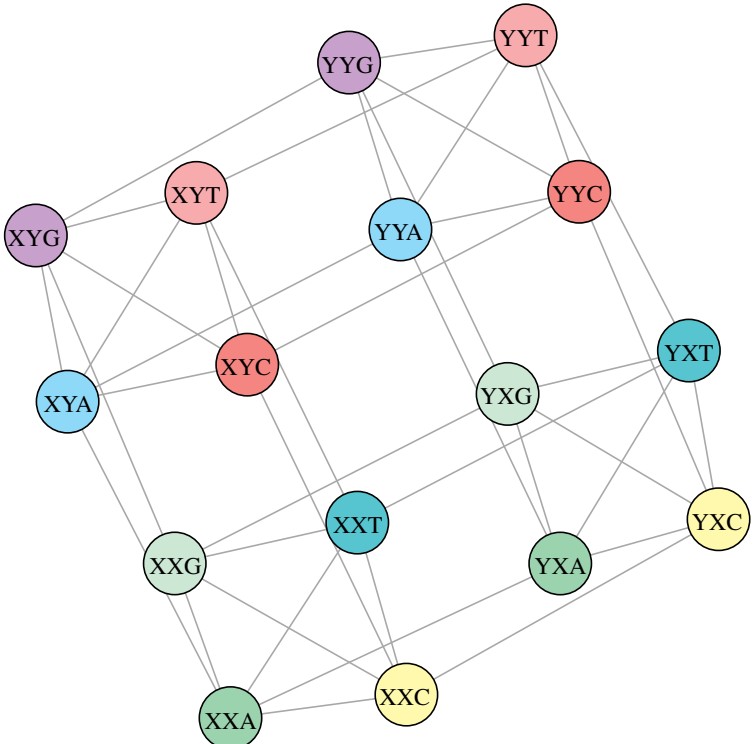

**Figure 4.** The graphical of the graph $G[B_{12}]$ which is the induced subgraph of the graph $G(V, E)$. Each node is a codon belonging to the set $B_{12}$, $B_{12} \subset V$, whereas the edges are incorporated from the set $E$.

first $n$ codons (nodes) of $G[B_{12}]$ ordered in the lexicographic order possess the optimal $k$-size conductance $\phi_k(G[B_{12}])$. In table 2, we present the list of all $G[B_{12}]$ nodes ordered in the lexicographic order. What is more, we evaluated also all possible values of $\phi_k(G[B_{12}])$ for the sets composed of the first $k$ nodes.

Similarly to the previous results, the best genetic code extensions, namely, $V_n^{'*}$, $1 \leq n \leq 48$ have the nested structure of the optimal codon blocks. It can be obtained by addition of the subsequent codons according to their lexicographic order. The new codons are selected from the subsequent sets of type $B_{ij}$ until the total number of included codons in a given set reaches 16.

## 3.5. The optimal extension of the standard genetic code up to 216 codons

The methodology presented in the previous section allows us to extend $C^1$ up to the $C^2$ set of codons involving 208 out of 216 possible codons. In order to extend the genetic code by over 208 meaningful codons, we must conduct a reasoning. From (3.13), we get that $C^2$ is the best $C^1$ extension involving 48 additional codons. What is more, applying proposition 2.4(g) we get that $C^2$ includes all non-standard codons, which are connected to $C^1$. As a result, the property 3.11 is not restrictive in the case when we try to extend $C^1$ in consecutive steps using definition 3.4 for $n > 48$. Therefore, similarly to the method presented in the previous section, we reformulate the problem of the optimal $C^1$ extension into the question of the optimal extension of the $C^2$ set.

**Definition 3.8.** The set $V_n^{''*}$, $C^2 \subset V_n^{''*}$ with exactly $1 \leq n \leq 8$ additional codons is optimal if

$$V_n^{''*} = \arg\min_{\{V_n'': V_n'' = C^2 \cup A_n''\}} E(V_n'', \overline{V_n''}), \tag{3.14}$$

where $A_n''$ possess one additional feature

$$A_n'' = \arg\max_{\{S: S \subseteq \overline{C^2}, |S| = n\}} E(C^2, S). \tag{3.15}$$

We introduce also a definition which is useful in describing the optimality of the genetic code extension.

**Table 2.** The codons composing the set $B_{12}$. They are arranged according to a selected lexicographic order. The values of the $k$-size conductance $\phi_k(G[B_{12}])$ calculated for the first $k$ codons in order are also presented.

| codon | $k$ | $\phi_k(G[B_{12}])$ |
|---|---|---|
| XXA | 1 | 1 |
| XXT | 2 | 0.1 |
| XXG | 3 | 0.6 |
| XXC | 4 | 0.4 |
| AYA | 5 | 0.440 |
| XYT | 6 | 0.4 |
| XYG | 7 | 0.314 |
| XYC | 8 | 0.2 |
| YXA | 9 | 0.244 |
| YXT | 10 | 0.240 |
| YXG | 11 | 0.2 |
| YXC | 12 | 0.133 |
| YYA | 13 | 0.138 |
| YYT | 14 | 0.114 |
| YYG | 15 | 0.067 |
| YYC | 16 | 0 |

**Definition 3.9.** Let us denote by $\mathcal{V}''_n$ a class of sets $C^2 \subset V''_n$ with $n \geq 1$ additional codons and $A''_n = V''_n \backslash C^2$ fulfils the property (3.15). Then

$$\mathcal{V}''_n = \{V''_n : V''_n = C^2 \cup A''_n\}$$

is a class of all possible extensions of the $C^2$ set with exactly $n$, $1 \leq n \leq 8$ additional codons.

Using definition 3.8 of optimality, we get the following characterization of the set $V''^*_n$.

**Theorem 3.10.** For every $1 \leq n \leq 8$, the following equation holds

$$V''^*_n = C^2 \cup A''_n,$$

where $V''^*_n \in \mathcal{V}''_n$ and $A''_n \subseteq B_{123}$ is optimal in terms of $\phi_k(G[B_{123}])$.

*Proof.* The proof of this theorem is an immediate consequence of proposition 2.4(g) and definition 2.11. ∎

Furthermore, the induced subgraph $G[B_{123}]$ (figure 5) can be also represented as a Cartesian product of graphs (proposition 2.9). Using again theorem 2.3 from [35], we obtain that the collection of the first $n$ codons of $G[B_{123}]$ taken in the lexicographic order possess the optimal $k$-size conductance $\phi_k(G[B_{123}])$. In table 3, we present the list of all $G[B_{123}]$ nodes taken in the selected lexicographic order. We evaluated also all possible values of the $k$-size conductance for the sets composed of the first $k$ nodes.

# 4. Discussion

The huge number of combinations of non-canonical and canonical bases in the creation of new codon groups means that the genetic code can be extended in various ways. Here, we propose the SGC extension in three steps consisting of the addition of codons including an increasing number of non-canonical bases. These steps extend the genetic code to 160 meaningful codons, then to 208 codons and finally to 216 codons. The extension of the SGC proposed by us is a general approach, which does not take into account properties of coded amino acids or other compounds associated with the newly added codons. We focused on the global structure of the code including arrangement of codons in groups (blocks) coding a given amino acid which differed usually in one codon position. The codons are added according to a fixed lexicographic order, which makes the EGC robust to changes causing the loss of genetic information.

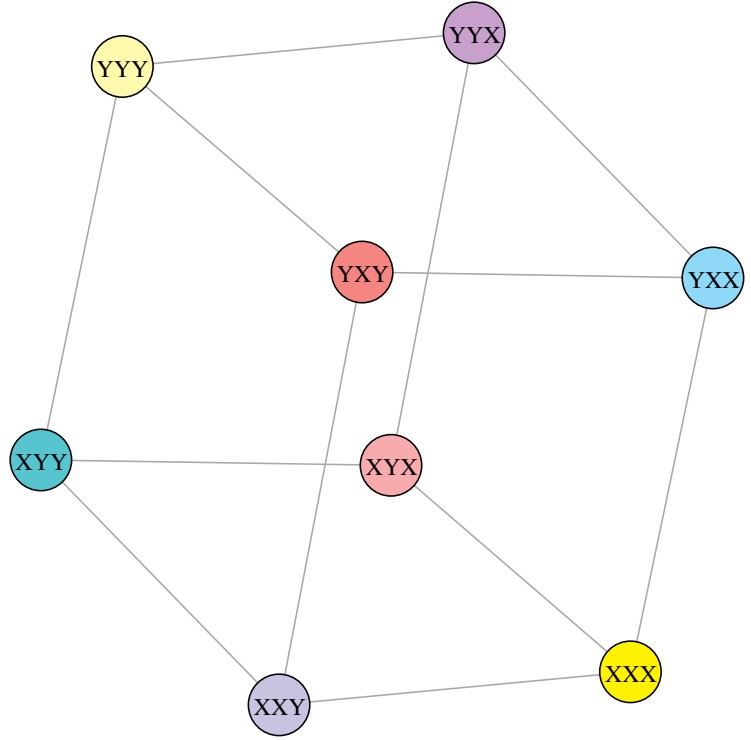

**Figure 5.** The graphical of the graph $G[B_{123}]$ which is the induced subgraph of the graph $G(V, E)$. Each node is a codon belonging to the set $B_{123}$, $B_{123} \subset V$, whereas the edges are incorporated from the set $E$.

**Table 3.** The sequence of codons composing the set $B_{123}$. They are ordered according to a selected lexicographic order. The values of the $k$-size conductance $\phi_k(G[B_{123}])$ calculated for the first $k$ codons in order are also presented.

| codon | $k$ | $\phi_k(G[B_{123}])$ |
|---|---|---|
| XXX | 1 | 1 |
| XXY | 2 | 0.667 |
| XYX | 3 | 0.556 |
| XYY | 4 | 0.333 |
| YXX | 5 | 0.333 |
| YXY | 6 | 0.222 |
| YYX | 7 | 0.143 |
| YYY | 8 | 0 |

This approach conforms the assumption of the adaptation hypothesis, which postulates that the SGC evolved to minimize harmful consequences of mutations or mistranslations of coded proteins [24,29,32,33]. The SGC turned out to be quite well optimized in this respect when compared with a sample of randomly generated codes [25,26,28,31,36] but the application of optimization algorithms revealed that the SGC is not perfectly optimized in this respect and more robust codes can be found [34,37–44]. The minimization of mutation errors is important from a biological point of view, because it protects organisms against losing genetic information. Then, the reduction of the mutational load seems favoured by biological systems and can occur directly at the level of the mutational pressure [45–49]. Nevertheless, in the global scale, the SGC shows a general tendency to error minimization [37,44], which is more exhibited by its alternative versions [50], evolved later. Therefore, the extension of the SGC according to this rule seems to be a natural consequence of its evolution.

Our approach assumes a stepwise extension of the code similarly to the gradual addition of new amino acids to the evolving primordial SGC, when they were produced by increasingly more complex biosynthetic pathways evolving in parallel [51–60]. The addition of amino acids was also driven by

the selection for the increasing diversity of amino acids [61–64] as well as decreasing disruption of already coded proteins and their composition [65]. The similar assumptions are included in our model, which assumes the minimization of differences between the newly added codons and those already defined in the code. The codons added in the first step contain, besides two canonical bases (N), only one non-canonical base (X), i.e. XNN, NXN and NNX, thus differing from the typical codons (NNN) in only one position. The next added codons include already two non-canonical bases, i.e. XXN, XNX and NXX. Finally, codons consisting exclusively of three non-canonical bases (XXX) can be used to extend the code. This method of codon addition causes the newly added codons to differ from the current ones in one point mutation.

Thanks to this gradual addition of new codons with assigned new amino acids, the whole system, i.e. an organism with the extended code, can have a quite high probability of surviving. The inclusion of new codons that differ in one mutation step between themselves and the canonical ones means that in the case of such mutation there is a small probability of undefined codons being generated, which could cause premature termination of translation of coded products and their non-functionality. Assuming that reverse mutations are more frequent, i.e. substitutions of a non-canonical base by a canonical one, the EGC can be reduced with time to the SGC. The stepwise addition can also give an organism time for adaptation and tuning the molecular processes to the new products. Moreover, it enables better monitoring and control of the organism's modification.

However, from an experimentalist point of view, such reversions would not be desirable because the modified system would revert to the original one. Therefore, we can imagine an alternative way of the genetic code extension by adding codons that cannot be mutated in a single step to the already defined codons in the code. To extend the SGC in this way, the first added codon sets should contain at least two non-canonical bases, i.e. XXX or XXN, XNX and NXX. Then, any single mutation of these codons would cause generation of undefined codons and organisms bearing such a mutation could be naturally eliminated from the whole population if the mutation is deleterious. Our model of the SGC extension can be upgraded to include properties of newly added amino acids or other compounds which are introduced into the code.

Data accessibility. All the data generated or analysed during this study are included in this published article.

Authors' contributions. Conceived and designed the study: P.B. and P.M. Wrote and corrected the paper: P.B., P.M., M.W. and D.M. Responded to reviewers: P.B. and P.M. All authors participated in the improvement of the manuscript and approved the final version.

Competing interests. The authors declare that they have no competing interests.

Funding. This work was supported by the National Science Centre, Poland (Narodowe Centrum Nauki, Polska) under grant no. 2017/27/N/NZ2/00403.

Acknowledgements. We are very grateful to two anonymous reviewers for their insightful comments and remarks, which significantly improved the manuscript.

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
