## [Reviewer comments · Royal Society Open Science]

Review History

RSOS-191384.R0 (Original submission)

Review form: Reviewer 1

Is the manuscript scientifically sound in its present form?

Yes

Are the interpretations and conclusions justified by the results?

Yes

Is the language acceptable?

Yes

Do you have any ethical concerns with this paper?

No

Have you any concerns about statistical analyses in this paper?

No

Recommendation?

Major revision is needed (please make suggestions in comments)

Comments to the Author(s)

The article under review discusses optimal extensions of the genetic code that use two non-canonical nucleotide bases to encode non-canonical amino acids. The criterion for the optimality of such an extension is the minimal influence of point mutations on the extended genetic code. The topic is very up-to-date and the optimization criterion is biologically meaningful. The article is well written and structured. However, in my opinion, there are several points that could be criticized:

1. Why does the article discuss the Standard Genetic Code (SGC) while the assignment of amino acids to single codons plays no role in its context?
2. The article is generally very technical and contains many technical definitions and notations, but all the tables and figures illustrating them are given at the end of the article while the necessary explanations for them are many pages before. Therefore, to understand the article, it would be helpful if Tables 1-3 and Figures 1-4 were not placed at the end of the article but at the places where the definitions are given. It would also be helpful if there were some practical examples, especially ones that show what the optimal sets A_k for certain (small) k could look like and that there can be more than one A_k for a certain k .
3. In Definition 4, k is used in a different meaning than the one in the definition of V_k , where k stands for the number of non-canonical codons. The article also uses Notation X , with or without indices, to denote very different objects. All that could be confusing.
4. In general, the number of notations is very big, which blurs the point made by the approach suggested. Perhaps, this should be reconsidered.
5. At the beginning of Section 1.2, it would be helpful if there was a reminder of what k_i $i=1,2,3$ means.
6. Strictly speaking, Proposition 3 only applies to $n=1$. $G[X_2]$ and $G[X_3]$ are only isomorphic to the given graph. Similarly, Proposition 4 only applies to $n=12$.
7. One of the results of the article is that, for instance, the extension XV_c is already optimal. Why should other variants be considered if the assignment of amino acids to single codons is not important for the article anyway, as already mentioned in Point 1?

Review form: Reviewer 2

Is the manuscript scientifically sound in its present form?

Yes

Are the interpretations and conclusions justified by the results?

Yes

Is the language acceptable?

Yes

Do you have any ethical concerns with this paper?

Yes

Have you any concerns about statistical analyses in this paper?

No

Recommendation?

Accept with minor revision (please list in comments)

Comments to the Author(s)

This is a technical but nevertheless interesting manuscript. Authors are a known group that operates also on the origin of the genetic code field. The only revision that I suggest consists of

the following. Authors have only to add, in the discussion, some sentences that justify the gradual addition of amino acids to the genetic code. After this revision, the manuscript could be accepted.

Decision letter (RSOS-191384.R0)

22-Nov-2019

Dear Dr Błażej,

The editors assigned to your paper ("Basic principles of the genetic code extension") have now received comments from reviewers. We would like you to revise your paper in accordance with the referee and Associate Editor suggestions which can be found below (not including confidential reports to the Editor). Please note this decision does not guarantee eventual acceptance.

Please submit a copy of your revised paper before 15-Dec-2019. Please note that the revision deadline will expire at 00.00am on this date. If we do not hear from you within this time then it will be assumed that the paper has been withdrawn. In exceptional circumstances, extensions may be possible if agreed with the Editorial Office in advance. We do not allow multiple rounds of revision so we urge you to make every effort to fully address all of the comments at this stage. If deemed necessary by the Editors, your manuscript will be sent back to one or more of the original reviewers for assessment. If the original reviewers are not available, we may invite new reviewers.

- Data accessibility

It is a condition of publication that all supporting data are made available either as supplementary information or preferably in a suitable permanent repository. The data accessibility section should state where the article's supporting data can be accessed. This section should also include details, where possible of where to access other relevant research materials such as statistical tools, protocols, software etc can be accessed. If the data have been deposited in an external repository this section should list the database, accession number and link to the DOI

for all data from the article that have been made publicly available. Data sets that have been deposited in an external repository and have a DOI should also be appropriately cited in the manuscript and included in the reference list.

If you wish to submit your supporting data or code to Dryad (<http://datadryad.org/>), or modify your current submission to dryad, please use the following link:
<http://datadryad.org/submit?journalID=RSOS&manu=RSOS-191384>

- **Competing interests**

- **Authors' contributions**

- **Acknowledgements**

- **Funding statement**

Kind regards,

on behalf of the Associate Editor and Professor Mark Chaplain (Subject Editor)
openscience@royalsociety.org

Associate Editor's comments:

Thank you kindly for submitting your manuscript to Royal Society Open Science.

Both Referee #1 and Referee #2 agree that your manuscript is well written and structured, and that your manuscript provides a useful contribution to the literature. However, as particularly

specified by Referee #1, there are a few points that need to be addressed before this manuscript is suitable for publication. Referee #2 suggested that some sentences that justify the gradual addition of amino acids to the genetic code be added to your manuscript.

Please ensure that you address each of the referees' comments while preparing your manuscript, and provide a point-by-point response on how you made these changes when you submit your revision. Please ensure to provide a tracked changes copy of your manuscript, which highlights the changes you have made.

Reviewers' Comments to Author:

Reviewer: 1

Comments to the Author(s)

The article under review discusses optimal extensions of the genetic code that use two non-canonical nucleotide bases to encode non-canonical amino acids. The criterion for the optimality of such an extension is the minimal influence of point mutations on the extended genetic code. The topic is very up-to-date and the optimization criterion is biologically meaningful. The article is well written and structured. However, in my opinion, there are several points that could be criticized:

1. Why does the article discuss the Standard Genetic Code (SGC) while the assignment of amino acids to single codons plays no role in its context?
2. The article is generally very technical and contains many technical definitions and notations, but all the tables and figures illustrating them are given at the end of the article while the necessary explanations for them are many pages before. Therefore, to understand the article, it would be helpful if Tables 1-3 and Figures 1-4 were not placed at the end of the article but at the places where the definitions are given. It would also be helpful if there were some practical examples, especially ones that show what the optimal sets A_k for certain (small) k could look like and that there can be more than one A_k for a certain k .
3. In Definition 4, k is used in a different meaning than the one in the definition of V_k , where k stands for the number of non-canonical codons. The article also uses Notation X , with or without indices, to denote very different objects. All that could be confusing.
4. In general, the number of notations is very big, which blurs the point made by the approach suggested. Perhaps, this should be reconsidered.
5. At the beginning of Section 1.2, it would be helpful if there was a reminder of what k_i $i=1,2,3$ means.
6. Strictly speaking, Proposition 3 only applies to $n=1$. $G[X_2]$ and $G[X_3]$ are only isomorphic to the given graph. Similarly, Proposition 4 only applies to $n=12$.
7. One of the results of the article is that, for instance, the extension XV_c is already optimal. Why should other variants be considered if the assignment of amino acids to single codons is not important for the article anyway, as already mentioned in Point 1?

Reviewer: 2

Comments to the Author(s)

This is a technical but nevertheless interesting manuscript. Authors are a known group that operates also on the origin of the genetic code field. The only revision that I suggest consists of the following. Authors have only to add, in the discussion, some sentences that justify the gradual addition of amino acids to the genetic code. After this revision, the manuscript could be accepted.

Author's Response to Decision Letter for (RSOS-191384.R0)

See Appendices A-B.

RSOS-191384.R1 (Revision)

Review form: Reviewer 1

Is the manuscript scientifically sound in its present form?

Yes

Are the interpretations and conclusions justified by the results?

Yes

Is the language acceptable?

Yes

Do you have any ethical concerns with this paper?

No

Have you any concerns about statistical analyses in this paper?

No

Recommendation?

Accept with minor revision (please list in comments)

Comments to the Author(s)

The authors have considered the comments of the reviewer. There remains one small change that should be made: In propositions 3 and 4 the notation $G[B_n]$ is used, where n is not defined.

Review form: Reviewer 2

Is the manuscript scientifically sound in its present form?

Yes

Are the interpretations and conclusions justified by the results?

Yes

Is the language acceptable?

Yes

Do you have any ethical concerns with this paper?

No

Have you any concerns about statistical analyses in this paper?

No

Recommendation?

Accept as is

Comments to the Author(s)

It is a good paper

Decision letter (RSOS-191384.R1)

02-Jan-2020

Dear Dr Błażej:

On behalf of the Editors, I am pleased to inform you that your Manuscript RSOS-191384.R1 entitled "Basic principles of the genetic code extension" has been accepted for publication in Royal Society Open Science subject to minor revision in accordance with the referee suggestions. Please find the referees' comments at the end of this email.

The reviewers and Subject Editor have recommended publication, but also suggest some minor revisions to your manuscript. Therefore, I invite you to respond to the comments and revise your manuscript.

- Ethics statement

- Data accessibility

<http://datadryad.org/submit?journalID=RSOS&manu=RSOS-191384.R1>

- Competing interests

- Authors' contributions

- Acknowledgements

- Funding statement

Because the schedule for publication is very tight, it is a condition of publication that you submit the revised version of your manuscript before 11-Jan-2020. Please note that the revision deadline will expire at 00.00am on this date. If you do not think you will be able to meet this date please let me know immediately.

Supplementary files will be published alongside the paper on the journal website and posted on the online figshare repository (<https://figshare.com>). The heading and legend provided for each

supplementary file during the submission process will be used to create the figshare page, so please ensure these are accurate and informative so that your files can be found in searches. Files on figshare will be made available approximately one week before the accompanying article so that the supplementary material can be attributed a unique DOI.

on behalf of Prof Mark Chaplain (Subject Editor)
openscience@royalsociety.org

Associate Editor Comments to Author:

It appears that the authors have largely resolved the queries of the referees, though a small change is required.

Reviewer comments to Author:

Reviewer: 2

Comments to the Author(s)

It is a good paper

Reviewer: 1

Comments to the Author(s)

The authors have considered the comments of the reviewer. There remains one small change that should be made: In propositions 3 and 4 the notation $G[B_n]$ is used, where n is not defined.

Author's Response to Decision Letter for (RSOS-191384.R1)

See Appendix C.

Decision letter (RSOS-191384.R2)

09-Jan-2020

Dear Dr Błażej,

It is a pleasure to accept your manuscript entitled "Basic principles of the genetic code extension" in its current form for publication in Royal Society Open Science.

Kind regards,

on behalf of the Associate Editor and Professor Mark Chaplain (Subject Editor)
openscience@royalsociety.org

Appendix A

Dear Editor,

Thank you for managing our manuscript and opportunity to improve our manuscript.

In the attachment you can find the revised version of manuscript RSOS-191384, entitled “Basic principles of the genetic code extension”, which we would like to resubmit for publication as an article to Royal Society Open Science.

We corrected the manuscript according to all Reviewers’ comments and remarks, which significantly improved the quality of the manuscript. Briefly, we corrected and simplified the notation as well as included the short discussion about the gradual addition of new codons and amino acids to the genetic code. We also included lacking sections.

We hope that the revised version of our manuscript meets the criteria for publishing in your excellent journal.

We look forward to hearing from you at your earliest convenience.

Yours sincerely,

Pawe Błażej

Appendix B

Reviewers' Comments to Author:

Reviewer: 1

Comments to the Authors

The article under review discusses optimal extensions of the genetic code that use two non-canonical nucleotide bases to encode non-canonical amino acids. The criterion for the optimality of such an extension is the minimal influence of point mutations on the extended genetic code. The topic is very up-to-date and the optimization criterion is biologically meaningful. The article is well written and structured. However, in my opinion, there are several points that could be criticized:

Authors' response

Thank you very much for insightful and helpful expertise about our manuscript. We regarded all your remarks in the revised version.

Comments to the Authors

Why does the article discuss the Standard Genetic Code (SGC) while the assignment of amino acids to single codons plays no role in its context?

Authors' response

We discussed the Standard Genetic Code (SGC) because the criterion of the SGC extension minimizes point mutations on the extended genetic code and this assumption well corresponds to the adaptation hypothesis postulating that the SGC evolved to minimize harmful effects of mutations or mistranslations of coded proteins. The consideration of amino acids and their properties gives only weights for these point mutations but the general idea remains the same. The extension of the SGC according to this rule seems to be a natural consequence of its evolution. Therefore, we think that this discussion is relevant and places our new findings in a wider context. We modified a sentence in the discussion to better justify our reasoning. We also added that the inclusion of properties of newly added amino acids or other compounds can upgrade our model of the SGC extension.

Comments to the Authors

The article is generally very technical and contains many technical definitions and notations, but all the tables and figures illustrating them are given at the end of the article while the necessary explanations for them are many pages before. Therefore, to understand the article, it would be

helpful if Tables 1-3 and Figures 1-4 were not placed at the end of the article but at the places where the definitions are given.

Authors' response

The current positions of figures and tables result from the format style which was used during the writing of the draft version of our manuscript. We hope that it is an initial format and the final version will included the correct placement of the figures and the tables. We will insist on the production editor to put them in the right place.

Comments to the Authors

It would also be helpful if there were some practical examples, especially ones that show what the optimal sets A_k for certain (small) k could look like and that there can be more than one A_k for a certain k .

Authors' response

We included the examples of these cases in Figure 2.

Comments to the Authors

In Definition 4, k is used in a different meaning than the one in the definition of V_k , where k stands for the number of non-canonical codons. The article also uses Notation X , with or without indices, to denote very different objects. All that could be confusing.

Authors' response

We introduced changes in the notation style to avoid the misunderstanding. We hope that these changes improve clarity of our reasoning.

Comments to the Authors

In general, the number of notations is very big, which blurs the point made by the approach suggested. Perhaps, this should be reconsidered.

Authors' response

We reduce the number of notations as much as possible to simplify the presentation.

Comments to the Authors

At the beginning of Section 1.2, it would be helpful if there was a reminder of what k_i $i=1,2,3$ means.

Authors' response

We included this information at the beginning of this section.

Comments to the Authors

Strictly speaking, Proposition 3 only applies to $n=1$. $G[X_2]$ and $G[X_3]$ are only isomorphic to the given graph. Similarly, Proposition 4 only applies to $n=12$.

Authors' response

The Reviewer is right. These propositions apply only to the case $n=1$ and $n=12$, respectively. Therefore, we clearly stated about it in these propositions.

Comments to the Authors

One of the results of the article is that, for instance, the extension XV_c is already optimal. Why should other variants be considered if the assignment of amino acids to single codons is not important for the article anyway, as already mentioned in Point 1?

Authors' response

In this paper, we proposed a stepwise procedure of non-standard codons addition. In this context XV_c is the optimal genetic code extension but only for 160 codons in total. However, we can extend this code to other variant, which include more codons and non-canonical bases. Although it is not easy to include in the code at present such a huge number of new codons due to limitation of molecular methods, we can hope that in the future it will be possible. Therefore, we presented also other possibilities. Moreover, the inclusion of other variants makes our model complete. In the discussion, we included other way of the SGC extension, which starts from variants other than the XV_c .

Reviewer: 2**Comments to the Author(s)**

This is a technical but nevertheless interesting manuscript. Authors are a known group that operates also on the origin of the genetic code field. The only revision that I suggest consists of the following. Authors have only to add, in the discussion, some sentences that justify the gradual addition of amino acids to the genetic code. After this revision, the manuscript could be accepted.

Authors' response

Thank you very much for this comment. We justified the gradual addition of new codons and amino acids to the genetic code in the discussion.

Appendix C

Reviewers' Comments to Author:

Associate Editor Comments to Author: It appears that the authors have largely resolved the queries of the referees, though a small change is required.

Authors' response: Thank you for managing our manuscript and opportunity to improve our paper.

Reviewer comments to Author:

Reviewer: 2

Comments to the Author(s): it is a good paper

Authors' response: Thank you very much. It is a pleasure to read such review.

Reviewer: 1

Comments to the Author(s): The authors have considered the comments of the reviewer. There remains one small change that should be made: In propositions 3 and 4 the notation $G[B_n]$ is used, where n is not defined.

Authors' response: Thank you very much. We corrected these typos.